# A Simple and Effective Framework for Pairwise Deep Metric Learning

## Abstract

Deep metric learning (DML) has received much attention in deep learning due to its wide applications in computer vision. Previous studies have focused on designing complicated losses and hard example mining methods, which are mostly heuristic and lack of theoretical understanding. In this paper, we cast DML as a simple pairwise binary classification problem that classifies a pair of examples as similar or dissimilar. It identifies the most critical issue in this problem—imbalanced data pairs. To tackle this issue, we propose a simple and effective framework to sample pairs in a batch of data for updating the model. The key to this framework is to define a robust loss for all pairs over a mini-batch of data, which is formulated by distributionally robust optimization. The flexibility in constructing the *uncertainty decision set* of the dual variable allows us to recover state-of-the-art complicated losses and also to induce novel variants. Empirical studies on several benchmark data sets demonstrate that our simple and effective method outperforms the state-of-the-art results.

## 1 Introduction

Metric Learning aims to learn a metric to measure the distance between examples that captures certain notion of human-defined similarity between examples. Deep metric learning (DML) has emerged as an effective approach for learning a metric by training a deep neural network. Simply speaking, a deep neural network can induce new feature embedding of examples and it is trained in such a way that the Euclidean distance between the induced feature embeddings of two similar examples shall be small and that between the induced feature embeddings of two dissimilar pairs shall be large. DML has been widely used in many tasks such as face recognition (Fan et al. (2017)), image retrieval (Chen & Deng (2019)), and classification (Qian et al. (2015); Li et al. (2019)).

However, unlike training a deep neural network by minimizing the classification error, training a deep neural network for metric learning is notoriously more difficult (Qian et al. (2018); Wang et al. (2017)). Many studies have attempted to address this challenge by focusing on several issues. The first issue is how to define a loss function over pairs of examples. A variety of loss functions have been proposed such as contrastive loss (Hadsell et al. (2006)), binomial deviance loss (Yi et al.), margin loss (Wu et al. (2017)), lifted-structure (LS) loss (Oh Song et al. (2016)), N-pair loss (Sohn (2016)), triplet loss (Schroff et al. (2015)), multi-similarity (MS) loss (Wang et al. (2019). The major difference between these pair-based losses lies at how the pairs interact with each other in a mini-batch. In simple pairwise loss such as binomial deviance loss, contrastive loss, and margin loss, pairs are regarded as independent of each other. In triplet loss, a positive pair only interacts with one negative pair. In N-pair loss, a positive pair interacts with all negative pairs. In LS loss and MS loss, a positive pair interacts with all positive pairs and all negative pairs. The trend is that the loss functions become increasingly complicated but are difficult to understand.

In parallel with the loss function, how to select informative pairs to construct the loss function has also received great attention. Traditional approaches that construct pairs or triplets over all examples before training suffer from prohibitive $O(N^2)$ or $O(N^3)$ sample complexity, where $N$ is the total number of examples. To tackle this issue, constructing pairs within a mini-batch is widely used in practice. Although it helps mitigate the computational and storage burden, slow convergence and model degeneration with inferior performance still commonly exist when using all pairs in a mini-batch to update the model. To combat this issue, various *pair mining* methods have been proposed to complement the design of loss function, such as hard (semi-hard) mining for triplet loss (Schroff

et al. (2015)), distance weighted sampling (DWS) for margin loss (Wu et al. (2017)), MS sampling for MS loss (Wang et al. (2019)). These sampling methods usually keep all positive (similar) pairs and select roughly the same order of negative (dissimilar) pairs according to some criterion.

Regardless of these great efforts, existing studies either fail to explain the most fundamental problem in DML or fail to propose most effective approach towards addressing the fundamental challenge. It is evident that the loss functions become more and more complicated. But it is still unclear why these complicated losses are effective and how does the pair mining methods affect the overall loss within a mini-batch. In this paper, we propose a novel effective solution to DML and bring new insights from the perspective of learning theory that can guide the discovery of new methods. Our philosophy is simple: casting the problem of DML into a simple pairwise classification problem and focusing on addressing the most critical issue, i.e., the sheer imbalance between positive pairs and negative pairs. To this end, we employ simple pairwise loss functions (e.g., margin loss, binomial deviance loss) and propose a flexible distributionally robust optimization (DRO) framework for defining a robust loss over pairs within a mini-batch. The idea of DRO is to assign different weights to different pairs that are optimized by maximizing the weighted loss over an uncertainty set for the distributional variable. The model is updated by stochastic gradient descent with stochastic gradients computed based on the sampled pairs according to the found optimal distributional variable.

The DRO framework allows us to (i) connect to advanced learning theories that already exhibit their power for imbalanced data, hence providing theoretical explanation for the proposed framework; (ii) to unify pair sampling and loss-based methods to provide a unified perspective for existing solutions; (iii) to induce simple and effective methods for DML, leading to state-of-the-art performance on several benchmark datasets. The contributions of our work are summarized as follows:

- We propose a general solution framework for DML, i.e., by defining a robust overall loss based on the DRO formulation and updating the model based on pairs sampled according to the optimized sampling probabilities. We provide theoretical justification of the proposed framework from the perspective of advanced learning theories.

- We show that the general DRO framework can recover existing methods based on complicated pair-based losses: LS loss and MS loss by specifying different uncertainty sets for the distributional variable in DRO. It verifies that our method is general and brings a unified perspective regarding pair sampling and complicated loss over all pairs within a batch.

- We also propose simple solutions under the general DRO framework for tackling DML. Experimental results show that our proposed variants of DRO framework outperform state-of-the-art methods on several benchmark datasets.

## 2 RELATED WORK

**Loss Design.** The loss function is usually defined over the similarities or distances between the induced feature embeddings of pairs. There are simple pairwise losses that simply regard DML as binary classification problem using averaged loss over pairs, e.g., contrastive loss, binomial loss, margin loss. It is notable that the binomial loss proposed in (Yi et al.) assigns asymmetric weights for positive and negative pairs, which can mitigate the issue of imbalance to certain degree. The principal of the newly designed complicated *pair-based* losses can be summarized as heuristically discovering specific kinds of relevant information between groups of pairs to boost the training. The key difference between these complicated losses lies at how to group the pairs. N-pair loss put one positive pair and all negative pairs together, Lifted-structure loss and MS-loss group all positive pairs together and all negative pairs together for each example. In contrast, our DRO framework employs simple pairwise loss but induce complicated overall loss in a systematic and interpretable way.

**Pair Mining/Pair Weighting.** Wu et al. (2017) points out that pair mining plays an important role in distance metric learning. Different pair mining methods have been proposed, including semi-hard sampling for triplet loss, distance weighted sampling (DWS) for margin loss, MS mining for MS losses. These pair mining methods aim to select the hard positive and negative pairs for each anchor. For instance, Schroff et al. (2015) selects the hard negative pairs whose distance is smaller than that between the positive pairs in triplets, Shi et al. (2016) selects the hardest positive pair whose distance is smaller than that of the nearest negative pair in a batch, and MS mining (Wang et al. (2019)) selects hard negative pairs whose distance is smaller than the largest distance between positive pairs and hard positive pairs whose distance is larger than the smallest distance between negative pairs at the same time. DWS method keeps all positive pairs but samples negative pairs

according to their distance distribution within a batch. The proposed DRO framework induce a pair sampling method by using the optimal distributional variables that defines the robust loss over pairs within a mini-batch. As a result, the sampling probabilities induced by our DRO framework is automatically adaptive to the *pair-based losses*. There are other works that study the problem from the perspective of pair weighting instead of pair sampling. For example, Yu et al. (2018) heuristically design exponential weights for the different pairs in a triplet, which is a special case of our DRO framework. Details are provided in the supplementary. However, since the quality of anchors varies very much, it may not be reasonable to sample the same number of pairs from all anchors.

**Imbalance Data Classification.** There are many studies in machine learning which have tackled the imbalanced issue. Commonly used tricks include over-sampling, under-sampling and cost-sensitive learning. However, these approaches do not take the differences between examples into account. Other effective approaches grounded on advanced learning theories include minimizing maximal losses (Shalev-Shwartz & Wexler, 2016), minimizing top-k losses (Fan et al., 2017) and minimizing variance-regularized losses (Namkoong & Duchi, 2017). However, these approaches are not efficient for deep learning with big data, which is a severe issue in DML. In contrast, the proposed DRO formulation is defined over a mini-batch of examples, which inherits the theoretical explanation from the literature and is much more efficient for DML. In addition, the induced loss by our DRO formulation include maximal loss, top-k loss and variance-regularized loss as special cases by specifying different uncertainty sets of the distributional variable.

## 3 DML As A DRO-Based Binary Classification Problem

In this section, we will first present a general framework for DML based on DRO with theoretical justification. We will then discuss three simple variants of the proposed framework and also show how the proposed framework recover existing complicated losses for DML.

**Preliminaries.** Let $\mathbf{x} \in \mathbb{R}^D$ denote an input data (e.g., image) and $f(\cdot; \theta) : \mathbb{R}^D \to \mathbb{R}^d$ denote the feature embedding function defined by a deep neural network parameterized by $\theta$. The central task in DML is to update the model parameter $\theta$ by leveraging pairs of similar and dissimilar examples. Following most existing works, at each iteration we will sample a mini-batch of examples denoted by $\{\mathbf{x}_1, ..., \mathbf{x}_B\}$. We can construct $B^2$ pairs between these examples [1], and let $y_{ij}$ denote the label of pairs, i.e., $y_{ij} = 1$ if the pair is similar (positive), and $y_{ij} = 0$ if the pair is dissimilar (negative). The label of pairs can be either defined by users or derived from the class label of individual examples. Existing works of DML follow the same paradigm for learning the deep neural network i.e., a loss function $F(\theta)$ is first defined over the pairs within a mini-batch and the model parameter $\theta$ is updated by gradient-based methods. Various gradient-based methods can be used, including stochastic gradient descent (SGD), stochastic momentum methods and adaptive gradient methods (e.g. Adam). Taking SGD as an example, the model parameter $\theta$ can be updated by $\theta \leftarrow \theta - \eta \nabla F(\theta)$, where $\eta$ denotes the learning rate. The focus here is to how to define the loss function $F(\theta)$ over all pairs within a mini-batch. As mentioned earlier, we will cast the problem as simple binary classification problem, i.e., classifying a pair into positive or negative. To this end, we use $l_{ij}(\theta) = l(f(\mathbf{x}_i; \theta), f(\mathbf{x}_j; \theta), y_{ij})$ denote the pairwise classification loss between $\mathbf{x}_i$ and $\mathbf{x}_j$ in the embedding space (e.g., margin loss Wu et al. (2017), binomial loss Yi et al.). A naive approach for DML is to use the averaged loss over all pairs, i.e., $F_{\text{avg}}(\theta) = \frac{1}{B^2} \sum_{i=1}^{B} \sum_{j=1}^{B} l_{ij}(\theta)$. However, this approach will suffer from the severe imbalanced issue, i.e., most pairs are negative pairs. The gradient of $F_{\text{avg}}$ will be dominated by that of negative pairs.

### 3.1 General DRO-Based Framework

To address the imbalanced pair issue, we propose a general DRO formulation to compute a robust loss. The formulation of our DRO-based loss over all pairs within a mini-batch is given below:

$$F(\theta) = \max_{\mathbf{p} \in \mathcal{U}} \{g(\theta, \mathbf{p}) := \sum_{i=1}^{B} \sum_{j=1}^{B} p_{ij} l_{ij}(\theta)\}, \quad (1)$$

where $\mathbf{p} \in \mathbb{R}_+^{B^2}$ is a non-negative vector with each element $p_{ij}$ representing a weight for an individual pair. $\mathcal{U} \subseteq \mathbb{R}^{B^2}$ denotes the decision set of $\mathbf{p}$, which encodes some prior knowledge about $\mathbf{p}$. In

---

[1]For simplicity, we consider all pairs including self-pair.

the literature of DRO Namkoong & Duchi (2017), $\mathbf{p}$ is interpreted as a probability vector such that $\sum_{ij} p_{ij} = 1$ called the distributional variable and $\mathcal{U}$ denotes the uncertainty set that specifies how $\mathbf{p}$ deviates from the uniform probabilities $(1/B^2, \ldots, 1/B^2)$. In next subsection, we will propose simple variants of the above general framework by specifying different constraints or regularizations for $\mathbf{p}$. Below, we will provide some theoretical evidences to justify the above framework.

To theoretically justify the above loss, we connect (1) to exiting works in machine learning by considering three different uncertainty sets for $\mathbf{p}$. First, we can consider a simple constraint $\mathcal{U} = \Delta = \{p_{ij} \geq 0, \sum_{ij} p_{ij} = 1\}$. As a result, $F(\theta) = \max_{ij} l_{ij}(\theta)$ becomes the maximal loss over all pairs. Shalev-Shwartz & Wexler (2016) shows that minimizing maximum loss is robust to imbalanced data distributions and also derives better generalization error for imbalanced data with a rare class. However, the maximal loss is more sensitive to outliers (Zhu et al., 2019). To address this issue, top-$K$ loss (Fan et al., 2017) and variance-regularized loss (Namkoong & Duchi, 2017) are proposed, which can be induced by the above DRO framework. If we set $\mathcal{U} = \{\sum_{ij} p_{ij} = 1, 0 \leq p_{ij} \leq 1/K\}$, $F$ will become the top-$K$ loss $F(\theta) = \frac{1}{K} \sum_{i=1}^{K} l_{[i]}(\theta)$, where $l_{[i]}(\theta)$ denotes the $i$-th largest loss over all pairs. If we set $\mathcal{U}_\phi = \{\mathbf{p} \in \Delta, D_\phi(\mathbf{p} \| \mathbf{1}/B^2) \leq \frac{\rho}{B^2}\}$, where $D_\phi(\mathbf{p} \| \mathbf{p}') = \int \phi(\frac{d\mathbf{p}}{d\mathbf{p}'}) d\mathbf{p}'$ is the $\phi$-divergence between two distributions $\mathbf{p}$ and $\mathbf{p}'$ with $\phi(t) = \frac{1}{2}(t-1)^2$, then the DRO-based loss becomes the variance-regularized loss under certain condition about the variance of the random loss, i.e., for a set of i.i.d random losses $\{\ell_1, \ldots, \ell_n\}(n = B^2)$ we could have

$$\sup_{\mathbf{p} \in \mathcal{U}_\phi} \sum_{i=1}^{n} p_i \ell_i = \frac{1}{n} \sum_{i=1}^{n} \ell_i + \sqrt{\frac{2\rho \mathrm{Var}_n(\ell)}{n}},$$

where $\mathrm{Var}_n(\ell)$ denotes the empirical variance of $\ell_1, \ldots, \ell_n$. We can see that the second term in R.H.S of the above equation involves the variance, which can play a role of regularization. The variance-regularized loss has been justified from advanced learning theory by Namkoong & Duchi (2017), and its promising performance for imbalanced data has been observed as well.

Before ending this subsection, we will discuss how to update the model parameter $\theta$ based on the robust loss $F(\theta)$ defined by (1). A simple approach is to find an optimal distributional variable $\mathbf{p}_*$ to (1) and then update $\theta$ according to the subgradient of weighted loss by $\partial_\theta g(\theta, \mathbf{p}^*) = \sum_{ij} p_{ij}^* \nabla l_{ij}(\theta)$, which is justified by the following lemma.

**Lemma 1** *Assume that $g$ is proper, lower-semicontinuous in $\theta$ and level-bounded in $\mathbf{p}$ locally uniformly in $\theta$. Then the subgradient $\partial F(\theta) \subset \bigcup_{\mathbf{p}^* \in P^*(\theta)} \partial_\theta g(\theta, \mathbf{p}^*)$, where $P^*(\theta)$ denotes the optimal solution set of the maximization problem in (1). Furthermore, when $l_{ij}(\theta)$ is smooth in $\theta$ and $P^*(\theta)$ is a singleton, i.e., $\mathbf{p}^* = \arg\max_p g(\theta, \mathbf{p})$ is unique, we have $\partial F(\theta) = \partial_\theta g(\theta, p^*)$.*

**Remark 1** *The above lemma can be proved by Theorem 10.13 in Rockafellar & Wets (2009). It shows that even if we may not directly compute $\partial F(\theta)$, our framework can at least obtain its superset $\partial_\theta g(\theta, \mathbf{p}^*)$. Particularly, if we have additional conditions, i.e., $l_{ij}(\theta)$ is smooth in $\theta$ and the optimal solution $\mathbf{p}^*$ is unique (considering our regularized formulation below), it theoretically guarantees that our framework exactly computes $\partial F(\theta)$.*

### 3.2 PROPOSED THREE VARIANTS OF OUR FRAMEWORK

In this subsection, we present three variants of our general framework. In order to contrast to other variants recovering existing complicated losses presented in next subsection, we introduce some notations and make some simplifications. For each example $\mathbf{x}_i$ that serves as an anchor data, let $\mathcal{P}_i = \{j | y_{ij} = 1, j \in [B]\}$ and $\mathcal{N}_i = \{j | y_{ij} = 0, j \in [B]\}$ denote the index sets of positive and negative pairs, respectively. Let $\mathcal{P} = \bigcup_{i=1}^{B} \mathcal{P}_i$ and $N = \bigcup_{j=1}^{B} \mathcal{N}_i$. We denote the cardinality of a set by $P = |\mathcal{P}|$. For simplicity, we let $P_i = |\mathcal{P}_i|$, $N_i = |\mathcal{N}_i|$, $P = |\mathcal{P}|$ and $N = |\mathcal{N}|$. Since zero losses usually do not contribute to the computation of the subgradient for updating the model, we can simply eliminate those examples for consideration.

The first variant is to simply select the top-$K$ pairs with $K$-largest losses, which is equivalent to the following DRO formulation:

$$\text{DRO-TopK:} \quad \max_{\mathbf{p}} \sum_{i=1}^{B} \sum_{j \in \mathcal{P}_i \cup \mathcal{N}_i} p_{ij} l_{ij}(\theta), \text{ s.t. } \sum_{i=1}^{B} \sum_{j \in \mathcal{P}_i \cup \mathcal{N}_i} p_{ij} = 1, 0 \leq p_{ij} \leq 1/K,$$

where $K$ is a hyper-parameter. The gradient of the robust loss can be simply computed by sorting the pairwise losses and computing the average of top-$K$ losses.

The second variant is a variant of the variance-regularized loss. Instead of specifying the uncertainty set $\mathcal{U}_\phi$, we use a regularization term for the ease of computation, which is defined by

$$\text{DRO-KL:} \quad \max_{\mathbf{p} \in \mathbb{R}_+^{P+N}} \sum_{i=1}^B \sum_{j \in \mathcal{P}_i \cup \mathcal{N}_i} p_{ij} l_{ij}(\theta) - \gamma D_{KL}(\mathbf{p}||\frac{1}{P+N}), \text{ s.t. } \sum_{i=1}^B \sum_{j \in \mathcal{P}_i \cup \mathcal{N}_i} p_{ij} = 1,$$

where $\gamma > 0$ is a hyper-parameter and $D_{KL}$ denotes the KL divergence between two probabilities. The optimal solution to $\mathbf{p}$ can be easily computed following Namkoong & Duchi (2016). It is notable that the optimal solution $\mathbf{p}^*$ is not necessarily sparse. Hence, computing $\sum_{ij} p_{ij}^* \nabla l_{ij}(\theta)$ needs to compute the gradient of pairwise loss for all pairs, which could be prohibitive in practice when the mini-batch size is large. To alleviate this issue, we can simply sample a subset of pairs according to probabilities in $\mathbf{p}^*$ and the compute the averaged gradient of these sampled pairs.

The third variant of our DRO framework is explicitly balancing the number of positive pairs and negative pairs by choosing top $K/2$ pairs for each class, which is denoted by DRO-TopK-PN:

$$\text{DRO-TopK-PN:} \quad \max_{\mathbf{p} \in \{0,1\}^{P+N}} \sum_{i=1}^B \sum_{j \in \mathcal{P}_i \cup \mathcal{N}_i} p_{ij} l_{ij}(\theta), \text{ s.t. } \sum_{i=1}^B \sum_{j \in \mathcal{P}_i} p_{ij} \leq \frac{K}{2}, \sum_{i=1}^B \sum_{j \in \mathcal{N}_i} p_{ij} \leq \frac{K}{2}.$$

For implementation, we can simply select $K/2$ positive pairs with largest losses and $K/2$ negative pairs with largest loss respectively, and compute averaged gradient of the pairwise losses of the selected pairs for updating the model parameter.

### 3.3 RECOVERING THE METHOD BASED ON SOTA PAIR-BASED LOSS

Next we show that proposed framework can recover the method based on SOTA complicated losses. With the induced feature vector $f(\mathbf{x}; \theta)$ normalized to have unit norm, we define the similarity of two samples as $S_{ij} := \langle f(\mathbf{x}_i; \theta), f(\mathbf{x}_j; \theta) \rangle$, where $\langle \cdot, \cdot \rangle$ denotes dot product. Specifically, we consider two SOTA loss functions, LS and MS loss, which are defined below:

$$\mathcal{L}_{MS} = \frac{1}{n} \sum_{i=1}^n \{ \frac{1}{\alpha} \log[1 + \sum_{k \in \mathcal{P}_i} e^{-\alpha(S_{ik} - \lambda)}] + \frac{1}{\beta} \log[1 + \sum_{k \in \mathcal{N}_i} e^{\beta(S_{ik} - \lambda)}] \} \tag{2}$$

$$\mathcal{L}_{LS} = \sum_{i=1}^n [\log \sum_{k \in \mathcal{P}_i} e^{\lambda - S_{ik}} + \log \sum_{k \in \mathcal{N}_i} e^{S_{ik} - \lambda}]_+. \tag{3}$$

where $\alpha, \beta, \lambda$ are hyper-parameters of these losses.

The key to our argument is that the gradient computed based on these losses can be exactly computed according to our DRO framework by choosing appropriate constrained set $\mathcal{U}$ and setting the pairwise loss as the margin loss. To this end, we first show the gradient based on the LS loss, which can be computed by (Wang et al., 2019):

$$\frac{\partial \mathcal{L}(S)}{\partial \theta} = \frac{\partial \mathcal{L}(S)}{\partial S} \cdot \frac{\partial S}{\partial \theta} = \sum_{i=1}^B \sum_{j=1}^B \frac{\partial \mathcal{L}(S)}{\partial S_{ij}} \cdot \frac{\partial S_{ij}}{\partial \theta} \tag{4}$$

which can be alternatively written as

$$\frac{\partial \mathcal{L}(S)}{\partial \theta} = \sum_{i=1}^B \Big( \sum_{j \in \mathcal{N}_i} w_{ij}^- \frac{\partial S_{ij}}{\partial \theta} - \sum_{j \in \mathcal{P}_i} w_{ij}^+ \frac{S_{ij}}{\partial \theta} \Big). \tag{5}$$

It can be shown that for LS loss, derivations are provided in the supplementary, we have

$$w_{ij}^+ = \frac{e^{\lambda - S_{ij}}}{\sum\limits_{k \in \mathcal{P}_i}^B e^{\lambda - S_{ik}}} = \frac{1}{\sum\limits_{k \in \mathcal{P}_i}^B e^{S_{ij} - S_{ik}}}, \quad w_{ij}^- = \frac{e^{S_{ij} - \lambda}}{\sum\limits_{k \in \mathcal{N}_i}^B e^{S_{ik} - \lambda}} = \frac{1}{\sum\limits_{k \in \mathcal{N}_i}^B e^{S_{ik} - S_{ij}}}. \tag{6}$$

To recover the gradient of the LS loss under our DRO framework, we employ the pairwise margin loss for $l_{ij}(\theta)$, i.e., $l_{ij}(\theta) = [m + y_{ij}(\lambda - S_{ij})]_+$, where $m$ and $\lambda$ are two hyper-parameters and $[a]_+ = \max\{0, a\}$. Assume that the margin parameter $m$ is sufficiently large such that $l_{ij}(\theta) > 0$ for all pairs. The key to deriving the same gradient of the LS loss under our framework is to group

distributional variables in $\mathbf{p}$ for the positive and negative pairs according to the anchor data. Let $\mathbf{p}_i^+ \in \mathbb{R}^{P_i}$ and $\mathbf{p}_i^- \in \mathbb{R}^{N_i}$ denote the corresponding vectors of positive and negative pairs for the anchor $\mathbf{x}_i$, respectively. Let us consider the following DRO formulation:

$$F(\theta) = \max_{\mathbf{p} \in \mathbb{R}_+^{P+N}} \sum_{i=1}^{B} \sum_{j \in \mathcal{P}_i \cup \mathcal{N}_i} p_{ij} l_{ij}(\theta) - \sum_{i=1}^{B} \left( \gamma_i^+ D_{KL}(\mathbf{p}_i^+ \| \frac{\mathbf{1}}{P_i}) + \gamma_i^- D_{KL}(\mathbf{p}_i^- \| \frac{\mathbf{1}}{N_i}) \right)$$

$$\text{s.t. } \sum_{j \in \mathcal{P}_i} p_{ij} = 1, \sum_{k \in \mathcal{N}_i} p_{ik} = 1, \text{ for } i \in [B], \tag{7}$$

where $\gamma_i^+ \geq 0$ and $\gamma_i^- \geq 0$ for $i \in [B]$ are hyper-parameters. we can easily derive the closed-form solution for $\mathbf{p}_*$, i.e., $p_{ij}^{+*} = \frac{1}{\sum_{k \in \mathcal{P}_i} e^{(S_{ij} - S_{ik})/\gamma_i^+}}$, and $p_{ij}^{-*} = \frac{1}{\sum_{k \in \mathcal{N}_i} e^{(S_{ik} - S_{ij})/\gamma_i^-}}$. Then computing the gradient of the robust loss with respect to $\theta$ by using the above optimal $\mathbf{p}^*$, we have:

$$\partial F(\theta) = \sum_{i=1}^{B} \left( \sum_{j \in \mathcal{N}_i} p_{ij}^{-*} \frac{\partial S_{ij}}{\partial \theta} - \sum_{j \in \mathcal{P}_i} p_{ij}^{+*} \frac{\partial S_{ij}}{\partial \theta} \right)$$

which exactly recover the gradient in (6) by setting $\gamma_i^+ = \gamma_i^- = 1$.

Finally, we can recover the gradient based on the MS loss in a very similar way. The difference is to add a pseudo positive pair and pseudo negative pair with 0 loss for each anchor $\mathbf{x}_i$, and augment each $\mathbf{p}_i^+$ and $\mathbf{p}_i^-$ by one additional dimension. The details are provided in the supplementary.

## 4 EXPERIMENTS

Our methods was implemented by Pytorch and using BN-Inception network (Ioffe & Szegedy (2015)) pre-trained on ImageNet ILSVRC (Russakovsky et al. (2015)) to fairly compare with other works. The same as (Wang et al. (2019)), a FC layer on the top of the model structure following the global pooling layer was added with randomly initialization for our task. Adam Optimizer with $1e^{-}5$ learning rate was used for all our experiments.

We verify our DRO framework on image retrieval task with three standard datasets, Cub-200-2011, Cars-196 and In-Shop. These three datasets are split according to the standard protocol. For Cub-200-2011, the first 100 classes with 5864 images are used for training, and the the other 100 classes with 5924 images are saved for testing. Cars-196 consists of 196 car models with 16,185 images. We use the first 98 classes with 8054 images for training and the remaining 98 classes with 8,131 images for testing. For In-Shop, 997 classes with 25882 images are used for training. The test set is further partitioned to a query set with 14218 images of 3985 classes, and a gallery set containing 3985 classes with 12612 images. Batches are constructed with the following rule: we first sample a certain number of classes and then randomly sample $M$ instances for each class. The standard recall@k evaluation metric is used in all our experiments, where $k = \{1, 2, 4, 8, 16, 32\}$ on Cub-200-2011 and Car-196, and $k = \{1, 10, 20, 30, 40, 50\}$ on In-Shop. We apply margin loss ($\mathcal{L}_M$) and binomial loss ($\mathcal{L}_B$, Yi et al.) as base losses for our DRO framework. $m$ is the margin in $\mathcal{L}_M$. $\lambda$ is the threshold for both $\mathcal{L}_M$ and $\mathcal{L}_B$. $\alpha$ and $\beta$ are hyperparameters in $\mathcal{L}_B$.

### 4.1 QUANTITATIVE RESULTS

In this experiment, we compare our DRO framework with other SOTA baselines on Cub-200-2011, Cars-196 and In-Shop, which includes Wang et al. (2019); Yu et al. (2018); Kim et al. (2018); Opitz et al. (2018); Ge (2018); Harwood et al. (2017); Wu et al. (2017); Oh Song et al. (2017). Among them, mining-based methods are Clusetring, HDC, Margin, Smart Mining and HDL. ABIER and ABE are ensemble methods. HAP2S_E and MS are sampling-based methods, which are highly related to our methods. For our DRO framework, we test all three variants which are proposed in section 3. We apply two loss functions, margin loss and binomial loss, respectively. Since DRO $\mathbf{p}$-sampling works on all pairs in a batch, the binomial variant may not directly apply to p-sampling. Thus, it makes totally five variants of our DRO framework, denoted by DRO-TopK$_M$, DRO-TopK$_B$, DRO-TopK-PN$_M$, DRO-TopK-PN$_B$ and DRO-KL$_M$, where the subscript $M$ and $B$ represent the variants of our framework using margin loss and binomial loss, respectively. We set embedding space dimension $d = 1024$. The batchsize is set $B = 80$ on Cub-200-2011 and Cars-196, $B = 640$ on In-Shop. $\gamma$ is tuned from the range $= \{0.1 : 0.2 : 0.9\}$ on all three

**Table 1:** Recall@$k$ on Cub-200-2011 and Cars-196

| Recall@$k$(%) | Cub-200-2011 | | | | | | Cars-196 | | | | | |
|---|---|---|---|---|---|---|---|---|---|---|---|---|
| | 1 | 2 | 4 | 8 | 16 | 32 | 1 | 2 | 4 | 8 | 16 | 32 |
| Clusetring(Oh Song et al. (2017)) | 48.2 | 61.4 | 71.8 | 81.9 | - | - | 58.1 | 70.6 | 80.3 | 87.8 | - | - |
| HDC(Oh Song et al. (2017)) | 53.6 | 65.7 | 77.0 | 85.6 | 91.5 | 95.5 | 73.7 | 83.2 | 89.5 | 93.8 | 96.7 | 98.4 |
| Margin(Wu et al. (2017)) | 63.6 | 74.4 | 83.1 | 90.0 | 94.2 | - | 79.6 | 86.5 | 91.9 | 95.1 | 97.3 | - |
| Smart Mining(Harwood et al. (2017)) | 49.8 | 62.3 | 74.1 | 83.3 | - | - | 64.7 | 76.2 | 84.2 | 90.2 | - | - |
| HDL(Ge (2018)) | 57.1 | 68.8 | 78.7 | 86.5 | 92.5 | 95.5 | 81.4 | 88.0 | 92.7 | 95.7 | 97.4 | **99.0** |
| ABIER(Opitz et al. (2018)) | 57.5 | 68.7 | 78.3 | 86.2 | 91.9 | 95.5 | 82.0 | 89.0 | 93.2 | 96.1 | 97.8 | 98.7 |
| ABE(Kim et al. (2018)) | 60.6 | 71.5 | 79.8 | 87.4 | - | - | 85.2 | 90.5 | 94.0 | 96.1 | - | - |
| HAP2S_E(Yu et al. (2018)) | 56.1 | 68.3 | 79.2 | 86.9 | - | - | 74.1 | 83.5 | 89.9 | 94.1 | - | - |
| MS(Wang et al. (2019)) | 65.7 | 77.0 | **86.3** | 91.3 | 94.8 | 97.0 | 84.1 | 90.4 | 94.0 | 96.5 | **98.0** | 98.9 |
| DRO-TopK$_M$(Ours) | **67.4** | **77.7** | 85.9 | **91.6** | **95.0** | **97.3** | **86.0** | **91.7** | **95.0** | **97.3** | **98.5** | **99.2** |
| DRO-TopK$_B$(Ours) | **68.1** | **78.4** | 86.0 | **91.4** | **95.1** | **97.6** | **85.4** | **91.0** | **94.2** | **96.5** | **98.0** | **99.0** |
| DRO-TopK-PN$_M$(Ours) | **67.3** | **77.6** | 85.7 | **91.2** | **95.0** | **97.7** | **86.1** | **91.7** | **95.1** | **97.1** | **98.4** | **99.1** |
| DRO-TopK-PN$_B$(Ours) | **67.6** | **77.9** | 86.0 | **91.8** | **95.2** | **97.7** | **86.2** | **91.7** | **95.8** | **97.4** | **98.6** | **99.3** |
| DRO-KL$_M$(Ours) | **67.7** | **78.0** | 86.1 | **91.8** | **95.6** | **97.8** | **86.4** | **91.9** | **95.4** | **97.5** | **98.7** | **99.3** |

datasets and $K$ is tuned from $\{160, 200, 240, 280\}$ on Cub-200-2011 and Cars-196, and selected from $\{640, 960, 1280, 1600, 1920\}$ on In-Shop.

Table 1 and 3 report the experiment results. We mark the best performer in bold in the corresponding evaluation measure on each column. For our framework, particularly, we mark those who outperform all other SOTA methods in bold. We can see that our five variants outperform other SOTA methods on recall@1 on all three datasets. Particularly on Cars-196, our five variants outperforms other SOTA methods on all recall@$k$ measures. On Cub-200-2011, DRO-TopK$_B$ achieves a higher recall@1 (improving 2.4 in recall@1) than the best SOTA, MS. On Cars-196, DRO-KL$_M$ has the best performance, which improves 2.3 and 1.2 in recall@1 compared to the best non-ensemble SOTA, MS, and the best ensemble SOTA, ABE. On In-Shop, DRO-TopK-PN$_M$ improves 1.6 in recall@1 compared to the best results among SOTA, MS. The above results verify 1) the effectiveness of our DRO sampling methods and 2) the flexibility of our DRO framework to adopt different losses.

## 4.2 ABLATION STUDY

### 4.2.1 COMPARISON WITH LS LOSS AND MS LOSS

**Table 2:** Recover of MS loss and LS loss on Cub-200-2011 and Cars-196

| Recall@$K$(%) | Cub-200-2011 | | | | | | Cars-196 | | | | | |
|---|---|---|---|---|---|---|---|---|---|---|---|---|
| | 1 | 2 | 4 | 8 | 16 | 32 | 1 | 2 | 4 | 8 | 16 | 32 |
| MS | 55.6 | 67.7 | 77.4 | 86.3 | 92.1 | 95.8 | 73.2 | 81.5 | 87.6 | 92.6 | - | - |
| LS | 56.8 | 67.9 | 77.5 | 85.6 | 91.2 | 95.2 | 69.7 | 79.3 | 86.2 | 91.1 | - | - |
| DRO-KL-G-$\gamma = 1$ | 56.4 | **68.3** | **78.9** | 86.3 | 91.7 | 95.8 | 70.5 | 79.8 | 86.6 | 91.6 | **94.9** | **97.1** |
| DRO-KL-G-$\gamma = 0.1$ | 56.8 | **68.7** | **79.0** | 86.6 | 92.1 | 95.9 | 72.5 | **81.9** | **88.1** | 92.3 | **95.4** | **97.3** |
| DRO-KL-G-$\gamma = 0.01$ | **57.0** | **69.4** | **79.9** | 87.0 | 92.3 | 95.9 | **73.1** | **82.2** | **88.8** | 93.4 | **96.2** | **98.0** |
| DRO-KL-G-$\gamma = 0.001$ | 56.7 | **68.5** | **79.0** | 87.3 | 92.6 | 96.0 | **75.0** | **83.4** | **89.5** | 93.7 | **96.6** | **98.3** |

In Section 3.3, we theoretically show that LS loss and MS loss can be viewed as special cases of our DRO framework. In this experiment, we aim to empirically demonstrate that our framework is general enough and recovers LS loss. Specifically, we would show 1) when $\gamma = 1$, our framework performs similarly to LS loss, as stated in Section 3.3, 2) our framework can be seen as a *generalized LS loss* by treating $\gamma$ as a hyper-parameter, and 3) our *generalized LS loss* outperforms MS loss, even though the performance of the ordinary LS loss is inferior to that of MS loss.

We adopt the set up of embedding dimension and batchsize in the ablation study of Wang et al. (2019), i.e., $d = 64$ and $B = 80$. Therefore, we report the existing results of MS and LS loss presented in Wang et al. (2019) on Cars-196. For Cub-200-2011 and In-Shop, we implement MS and LS loss according to Wang et al. (2019). Following Wang et al. (2019), we set $\alpha = 2, \beta = 50$ for MS loss. For our DRO framework, we apply grouping to $\mathbf{p}$ by equation (7), and denote this variant of DRO framework as DRO-KL-G. We set $\gamma_i^+ = \gamma_i^- = \gamma = \{1, 0.1, 0.01, 0.001\}, i \in [B]$ for DRO-KL-G , $m = 0.2$ for the margin loss, and $\lambda = 0.5$ for all three losses (MS, LS and margin loss). As the pairs with zero loss will not contribute to the updates of model but affect the calculation of $\mathbf{p}$ in DRO framework, we remove the pairs with zero loss to further promotes training.

Table 2 and 3 show experiment results on Cub-200-2011, Cars-196 and In-Shop, respectively. As can be seen, the performance of MS loss is better than LS loss on three datasets, particularly on Cars-196, which also verifies the results of ablation study in Wang et al. (2019). When $\gamma = 1$, our method

performs similarly to LS loss, which verifies that our method recovers LS loss. Furthermore, when we treat $\gamma$ as a hyper-parameter (especially $\gamma = 0.001$) and regard our framework as *generalized LS loss*, our method obtain improved performance compared to the ordinary LS loss. Lastly, even if MS loss exploits pseudo positive and negative pairs, our *generalized LS loss* outperforms MS loss.

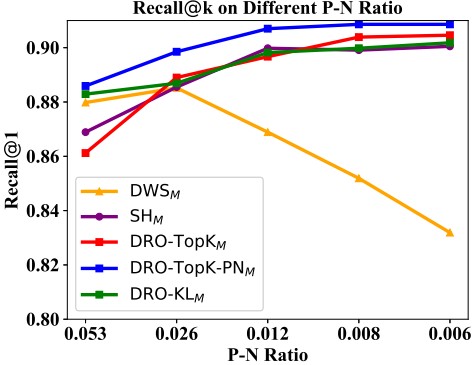

**Figure 1:** Recall vs Imbalance Ratio

**Table 3:** Recall@$k$ on In-Shop

| Recall@K | 1 | 10 | 20 | 30 | 40 | 50 |
|---|---|---|---|---|---|---|
| FashionNet(Liu et al. (2016)) | 53.7 | 73.0 | 76.0 | 77.0 | 79.0 | 80.0 |
| HDC(Oh Song et al. (2017)) | 62.1 | 84.9 | 89.0 | 91.2 | 92.3 | 93.1 |
| HDL(Ge (2018)) | 80.9 | 94.3 | 95.8 | 97.2 | 97.4 | 97.8 |
| ABIER(Opitz et al. (2018)) | 83.1 | 95.1 | 96.9 | 97.5 | 97.8 | 98.0 |
| ABE(Yu et al. (2018)) | 87.3 | 96.7 | 97.9 | 98.2 | 98.5 | 98.7 |
| MS(Wang et al. (2019)) | 89.7 | 97.9 | 98.5 | **98.8** | **99.1** | **99.2** |
| DRO-TopK$_M$(Ours) | **91.0** | **98.1** | **98.7** | 99.0 | **99.1** | **99.2** |
| DRO-TopK$_B$(Ours) | 90.7 | 97.7 | 98.4 | **98.8** | 99.0 | 99.1 |
| DRO-TopK-PN$_M$(Ours) | **91.3** | 98.0 | **98.7** | **98.9** | **99.1** | **99.2** |
| DRO-TopK-PN$_B$(Ours) | **91.1** | **98.1** | **98.6** | **98.8** | 99.0 | **99.2** |
| DRO-KL$_M$(Ours) | 90.8 | 98.0 | **98.6** | 99.0 | **99.1** | **99.2** |

**Table 4:** Recover of MS loss and LS loss on In-Shop

| Recall@K(%) | 1 | 10 | 20 | 30 | 40 | 50 |
|---|---|---|---|---|---|---|
| MS | 79.8 | 94.9 | 96.8 | 97.6 | 97.9 | 98.3 |
| LS | 82.6 | 94.1 | 95.6 | 96.4 | 96.9 | 97.4 |
| DRO-KL-G-$\gamma = 1$ | 84.8 | 95.9 | 97.3 | 97.9 | 98.2 | 98.5 |
| DRO-KL-G-$\gamma = 0.1$ | 85.1 | 96.1 | 97.5 | 98.0 | 98.3 | 98.5 |
| DRO-KL-G-$\gamma = 0.01$ | 85.8 | 96.2 | 97.9 | 97.8 | 98.2 | 98.4 |
| DRO-KL-G-$\gamma = 0.001$ | 85.7 | 96.1 | 97.4 | 97.9 | 98.2 | 98.5 |

### 4.2.2 CAPACITY TO HANDLE PAIR IMBALANCE.

In this experiment, we compare our DRO framework with different sampling methods, i.e., semi-hard (SH) and DWS, in terms of sensitivity to the imbalance ratio. By setting different batch-sizes $B \in \{80, 160, 320, 480, 640\}$, we have different positive-negative pair ratios $|\mathcal{P}| : |\mathcal{N}| \in \{0.053, 0.026, 0.012, 0.008, 0.006\}$. For all methods, we apply margin loss and set $M = 5$ for each class and embedding space dimension $d = 1024$. SH mining is originally designed for triplet loss. Since there is no straightforward choice for the positive pair, we use $\lambda$ as the upper bound to simulate the similarity of the positive pair in triplet loss. For DWS, we follow the parameter setting in the original paper (Wu et al. (2017)). We apply margin loss in the proposed three variants our DRO framework, which are denoted by DRO-TopK$_M$, DRO-TopK-PN$_M$ and DRO-KL$_M$, respectively. We set $K = 2 * B$ both for DRO-TopK$_M$, DRO-TopK-PN$_M$. We evaluate recall@1 of all methods and report results in Figure 1.

Figure 1 shows that the DWS has better performance when the positive-negative pair ratio is relatively large, and encounters a sharp decrease in recall@1 when the ratio decreases. Other four methods obtain better performance when the positive-negative ratio increases. Among them, DRO-TopK$_M$ and DRO-KL$_M$ have similar performance, with SH on all positive-negative pair ratios, while they perform slightly better than SH when the positive-negative ratios are small. DRO-TopK-PN$_M$ constantly outperforms all other methods. The reason why DWS performs poorly when the positive-negative pair ratio is small may be that DWS aims to sample pairs uniformly in terms of distance (Wu et al. (2017)), while our DRO framework and SH focus more on hard pairs. To sum up, our framework achieves very competitive performance against SOTA methods, and maintains increasing recall@1 as the positive-negative ratio increases. These two observations together demonstrate the effectiveness of our DRO framework to handle pair imbalance.

### 4.2.3 SENSITIVITY OF $K$ IN TOP-K

As we mentioned in section 1, selecting too many pairs within a batch will leads to poor performance of the model. On the other hand, when the number of selected pairs is too small, the model would be sensitive to outliers. In this experiment, we study the sensitivity of $K$ in our DRO framework–how the performance of our DRO framework is affected by the value of $K$. We set the batchsize $B = 640$ and $M = 5$, which makes the number of positive pairs $|\mathcal{P}| = 1280$ and the number of negative pairs $|\mathcal{N}| = 198080$. We set $K$ from the range $\{640, 960, 1280, 1600, 1920, 2560\}$ and evaluate recall@$k$ for models trained by different $K$. We choose the above range of $K$ according to the number of pairs selected by DWS and SH in Section 4.2.2 (both selects 2560 pairs roughly).

Figure 2 illustrates how different values of $K$ affect recall@$k$ on In-Shop. We can see that, DRO-TopK$_M$ performs best when $K = 1280$ and recall@$k$ is stable on the entire range of $K$. Our DRO framework is not sensitive to $K$ when $K$ is in a reasonably large range.

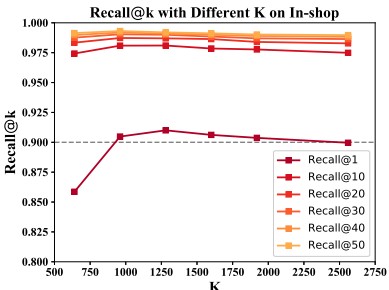
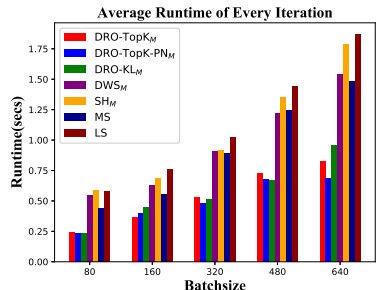

**Figure 2:** The effects of K on recall@$k$ on In-Shop    **Figure 3:** Average running time of every iteration

### 4.2.4 RUNTIME COMPARISON

Next, we compare the running time of our proposed three variants of our DRO framework with different pair mining methods, MS and LS losses on In-shop. Our experiments conducted on eight GTX1080Ti GPU. The embedding dimension $d = 1024$, and results are compared under different batchsize $B = \{80, 160, 320, 480, 640\}$. The same as previous experiments, we set $K = 2 * B$ both for DRO-TopK$_M$ and DRO-TopK-PN$_M$. $\gamma = 0.1$ for DRO-KL$_M$. SH is implemented according to the paper Schroff et al. (2015), Wu et al. (2017). DWS and MS are implemented based on the code provided by the author. LS loss is implemented following the code provided by Wang et al. (2019).

Figure 3 reports the average running time of each iteration on 200 epochs. We can see that all of three proposed variants of DRO framework run faster than other *anchor-based* mining methods and losses. For all of our three variants, pairs are selected directly from all the pairs, while additional cost is required to select pairs anchor by anchor in other methods. LS loss is slower than MS loss, because MS mining is applied to MS loss, which would reduce the number of pairs for computing subgradients when updating the model. For DWS, the distance distribution of negative pairs is only calculated once for each iteration. It thus only needs to select pairs according to the pre-computed distribution for each anchor. In contrast, SH requires to compare negative pairs with the lower and upper bound of an interval at each iteration for each anchor, which increases the computational burden. It can be the reason why SH is slower than DWS.

## 5 CONCLUSION

In this paper, we cast DML as a simple pairwise binary classification problem and formulate it as a DRO framework. Compared to existing DML methods that leverage all pairs in a batch or employ heuristic approaches to sample pairs, our DRO framework constructs a robust loss to sample informative pairs, which also comes with theoretical justification from the perspective of learning theory. Our framework is general since it can include many novel designs in its uncertainty decision set. Its flexibility allows us to recover the state-of-the-art loss functions. Experiments show that our framework outperforms the state-of-the-art DML methods on benchmark datasets. We also empirically demonstrate that our framework is efficient, general and flexible.

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

## 6 SUPPLEMENTARY

### 6.1 DERIVATION OF RECOVER SOTA LOSSES

In this section, we show how our DRO framework recovers SOTA loss functions, LS loss and MS loss. Their definitions are as follows, respectively.

$$\mathcal{L}_{LS} = \sum_{i=1}^{B}[\log \sum_{k \in P_i} e^{\lambda - S_{ik}} + \log \sum_{k \in N_i} e^{S_{ik} - \lambda}]_+, \tag{8}$$

where $\lambda$ is the margin hyper-parameter.

$$\mathcal{L}_{MS} = \frac{1}{n}\sum_{i=1}^{n}\{\frac{1}{\alpha}\log[1 + \sum_{k \in P_i} e^{-\alpha(S_{ik} - \lambda)}] + \frac{1}{\beta}\log[1 + \sum_{k \in N_i} e^{\beta(S_{ik} - \lambda)}]\} \tag{9}$$

where $\lambda$ is the margin hyper-parameter and $\alpha$ and $\beta$ are coefficient hyper-parameters.

### 6.1.1 LS LOSS UNDER OUR FRAMEWORK

Recall that the objective function is decomposable in terms of $\mathbf{p}_i = [\mathbf{p}_i^+, \mathbf{p}_i^-] \in \mathbb{R}^{P_i + N_i}$. We denote the $p_{ij} = p_{ij}^+$ when $j \in \mathcal{P}_i$, and $p_{ij} = p_{ij}^-$ when $j \in \mathcal{N}_i$ for simplicity. The Lagrangian function of (7) can be represented as:

$$\mathcal{L}(\mathbf{p}, \mathbf{v}) = \sum_{i=1}^{B}\mathcal{L}(\mathbf{p}_i, v_i^+, v_i^-), \tag{10}$$

where

$$\begin{aligned}\mathcal{L}(\mathbf{p}_i, v_i^+, v_i^-) = &-\sum_{j \in \mathcal{P}_i \cup \mathcal{N}_i} p_{ij}l_{ij}(\theta) + \gamma_i^+ D_{KL}(\mathcal{P}_i||\frac{\mathbf{1}}{|\mathcal{P}_i|}) + \gamma_i^- D_{KL}(\mathbf{p}_i^-||\frac{\mathbf{1}}{|\mathcal{N}_i|}) \\ &+ v_i^+(\sum_{j \in \mathcal{P}_i} p_{ij} - 1) + v_i^-(\sum_{j \in \mathcal{N}_i} p_{ij} - 1).\end{aligned} \tag{11}$$

According to KKT conditions, $v_i^{+*}$ and $v_i^{-*}$ are the optimal solutions of the dual function, and $\mathbf{p}_i^*$ is the optimal solution of the primal problem (7), if and only if

$$\frac{d\mathcal{L}}{d\mathbf{p}_i^*} = 0, \tag{12}$$

$$v_i^{+*}\Big(\sum_{j \in \mathcal{P}_i} p_{ij}^{+*} - 1\Big) + v_i^{-*}\Big(\sum_{j \in \mathcal{N}_i} p_{ij}^{-*} - 1\Big) = 0. \tag{13}$$

We first derive $\mathbf{p}_i^*$ in terms of $v_i^+, v_i^-$ using equation (12), i.e.:

$$\frac{d\mathcal{L}}{d\mathbf{p}_i^*} = -\mathbf{l}_i(\theta) + \gamma_i^+ \log(n\mathbf{p}_i^{+*}) + \gamma_i^- \log(n\mathbf{p}_i^{-*}) + \gamma_i^+ \mathbf{1}^+ + \gamma_i^- \mathbf{1}^- + v_i^+ \mathbf{1}^+ + v_i^- \mathbf{1}^- = 0 \tag{14}$$

where $\mathbf{l}_i(\theta) = \{l_{i1}(\theta), \cdots, l_{i|p_i^+ \cup N_i^+|}(\theta)\}, \mathbf{1}^+ \in R^{|\mathcal{P}_i|}, \mathbf{1}^- \in R^{|\mathcal{N}_i|}$. Then the closed form of $\mathbf{p}_i^*$ for positive pairs and negative pairs can be written as follows

$$p_{ij}^{+*} = \frac{1}{|\mathcal{P}_i|}e^{\frac{l_{ij}(\theta)-v_i^{+*}}{\gamma_i^+}-1}, p_{ij}^{-*} = \frac{1}{|\mathcal{N}_i|}e^{\frac{l_{ij}(\theta)-v_i^{-*}}{\gamma_i^-}-1}. \tag{15}$$

Substitute $p_{ij}^{+*}, p_{ij}^{-*}$ into equation (13), which means $v_i^{+*}$ and $v_i^{-*}$ need to satisfy:

$$v_i^{+*}\Big(\sum_{j \in \mathcal{P}_i} \frac{1}{|\mathcal{P}_i|}e^{\frac{l_{ij}(\theta)-v_i^{+*}}{\gamma_i^+}-1} - 1\Big) + v_i^{-*}\Big(\sum_{j \in \mathcal{N}_i} \frac{1}{|\mathcal{N}_i|}e^{\frac{l_{ij}(\theta)-v_i^{-*}}{\gamma_i^-}-1} - 1\Big) = 0. \tag{16}$$

Even though equal (16) also equals to 0 when $v_i^{+*} = 0$ or $v_i^{-*} = 0$, or $v_i^{+*} = v_i^{-*} = 0$, but the corresponding optimal solution $\mathbf{p}_i^*$ will not meet the equality constraints, i.e., $\sum_{j \in \mathcal{P}_i} p_{ij}^{+*} = 1$ and $\sum_{j \in \mathcal{P}_i} p_{ij}^{+*} = 1$, in the original formulation (7). Therefore, we only have

$$\sum_{j \in \mathcal{P}_i} \frac{1}{|\mathcal{P}_i|}e^{\frac{l_{ij}(\theta)-v_i^{+*}}{\gamma_i^+}-1} = 1, \quad \sum_{j \in \mathcal{N}_i} \frac{1}{|\mathcal{N}_i|}e^{\frac{l_{ij}(\theta)-v_i^{-*}}{\gamma_i^-}-1} = 1. \tag{17}$$

Then from equation (17), we can get

$$v_i^{+*} = \gamma_i^+ \log\Big(\sum_{j \in \mathcal{P}_i} \frac{1}{|\mathcal{P}_i|}e^{\frac{l_{ij}(\theta)}{\gamma}-1}\Big), \quad v_i^{-*} = \gamma_i^- \log\Big(\sum_{j \in \mathcal{N}_i} \frac{1}{|\mathcal{N}_i^-|}e^{\frac{l_{ij}(\theta)}{\gamma}-1}\Big). \tag{18}$$

Plugging them into (15) and apply margin loss as the base loss function for each pair, $l_{ij}(\theta) = [m + y_{ij}(\lambda - S_{ij})]_+$, we successfully derive the weighting representation of LS loss:

$$p_{ij}^{+*} = \frac{e^{\frac{l_{ij}(\theta)}{\gamma}}}{\sum_{k \in \mathcal{P}_i} e^{\frac{l_{ik}(\theta)}{\gamma}}} \overset{y_{ij}=1}{=} \frac{e^{\frac{[m+(\lambda-S_{ij})]_+}{\gamma}}}{\sum_{k \in \mathcal{P}_i} e^{\frac{[\alpha+[\lambda-S_{ik})]_+}{\gamma}}} = \frac{1}{\sum_{k \in \mathcal{P}_i} e^{\frac{S_{ij}-S_{ik}}{\gamma}}}$$

$$p_{ij}^{-*} = \frac{e^{\frac{l_{ij}(\theta)}{\gamma}}}{\sum_{k \in \mathcal{N}_i} e^{\frac{l_{ik}(\theta)}{\gamma}}} \overset{y_{ij}=-1}{=} \frac{e^{\big(\frac{[m+(S_{ij}-\beta)]_+}{\gamma}\big)}}{\sum_{k \in \mathcal{N}_i} e^{\frac{[\alpha+(S_{ik}-\beta)]_+}{\gamma}}} = \frac{1}{\sum_{k \in \mathcal{N}_i} e^{\frac{S_{ik}-S_{ij}}{\gamma}}} \tag{19}$$

Thus, when updating the model parameter $\theta$, we are going to minimize the following objective function:

$$g(\theta, \mathbf{p}^*) = \sum_{i=1}^{B} \sum_{j \in \mathcal{P}_i \cup \mathcal{N}_i} p_{ij}^* l_{ij}(\theta)$$

$$= \sum_{i=1}^{B} \Big(\sum_{j \in \mathcal{P}_i} p_{ij}^{+*} l_{ij}(\theta) + \sum_{j \in \mathcal{N}_i} p_{ij}^{-*} l_{ij}(\theta)\Big) \tag{20}$$

Taking the gradients to equation 20 in terms of $\theta$, we can get:

$$\frac{\partial g(\theta, \mathbf{p}^*)}{\partial \theta} = \sum_{i=1}^{B} \Big( \sum_{j \in \mathcal{P}_i} -p_{ij}^{+*} \frac{\partial S_{ij}}{\partial \theta} + \sum_{j \in \mathcal{N}_i} p_{ij}^{-*} \frac{\partial S_{ij}}{\partial \theta} \Big) \tag{21}$$

Similarly, we take gradients to the $\mathcal{L}_{LS}$ loss function (8):

$$\frac{\partial \mathcal{L}_{LS}}{\partial \theta} = \sum_{i=1}^{B} \Big( \sum_{j \in \mathcal{P}_i} \frac{-1}{\sum_{k \in \mathcal{P}_i}^{B} e^{S_{ij} - S_{ik}}} \frac{\partial S_{ij}}{\partial \theta} + \sum_{j \in \mathcal{N}_i} \frac{1}{\sum_{k \in \mathcal{N}_i}^{B} e^{S_{ik} - S_{ij}}} \frac{\partial S_{ij}}{\partial \theta} \Big) \tag{22}$$

By substituting equation (19) into equation (26), we can see that equation (22) and (26) are equivalent. This shows our DRO framework successfully recovers the LS loss by setting the uncertainty decision set $\mathcal{U}$ in equation (7).

### 6.1.2 MS Loss under Our Framework

MS loss, a combination of binomial loss and LS loss, can also be formulated into our DRO framework. LS loss only considers the lifted structure between pairs, while binomial loss focusing on the intrinsic property of an independent pair while encoding the pair class information. To recover MS loss under our framework, we re-define $\mathbf{p} \in [0,1]^{P^+ + N^+ + 2B}$ by adding one more element to $\mathcal{P}_i$ and $\mathbf{p}_i^-$. Therefore, now we have $\mathbf{p}_i^+ \in [0,1]^{P_i^+ + 1}$ and $\mathbf{p}_i^- \in [0,1]^{N_i^+ + 1}$, where the newly added element corresponds to a zero loss, and thus does not contribute to the computation of overall loss. Then based on the formulation of LS loss, the formulation of MS loss can be written as:

$$\max_{\mathbf{p} \in [0,1]^{P^+ + N^+ + 2B}} \sum_{i=1}^{B} \Big( \sum_{j \in \mathcal{P}_i \cup \mathcal{N}_i} p_{ij} l_{ij}(\theta) - \gamma_i^+ D_{KL}(\mathcal{P}_i \| \frac{1}{|\mathcal{P}_i| + 1}) - \gamma_i^- D_{KL}(\mathbf{p}_i^- \| \frac{1}{|\mathcal{N}_i| + 1}) \Big)$$

$$\text{s.t.} \sum_{j \in \mathcal{P}_i + 1} p_{ij} = 1, \sum_{k \in \mathcal{N}_i + 1} p_{ik} = 1, i \in [B]. \tag{23}$$

Note that

$$\sum_{j \in \mathcal{P}_i \cup \mathcal{N}_i} p_{ij} l_{ij}(\theta) + p_{i,P_i^+}^+ \cdot 0 + p_{i,N_i^+}^- \cdot 0 = \sum_{j \in \mathcal{P}_i + 1 \cup \mathcal{N}_i + 1} p_{ij} l_{ij}(\theta), \forall i \in [B]. \tag{24}$$

As the analysis of LS loss, we can also obtain the representation of MS loss under our DRO framework from formulation (23), i.e.,

$$p_{ij}^{+*} = \frac{e^{\frac{l_{ij}(\theta)}{\gamma_i^+}}}{e^{\frac{l_{i,|\mathcal{P}_i|+1}}{\gamma_i^+}} + \sum_{k \in P_i} e^{\frac{l_{ik}}{\gamma_i^+}}} \overset{y_{ij}=1}{=} \frac{e^{\frac{[m+(\lambda - S_{ij})]_+}{\gamma_i^+}}}{1 + \sum_{k \in \mathcal{P}_i} e^{\frac{[m+(\lambda - S_{ik})]_+}{\gamma_i^+}}} = \frac{1}{e^{\frac{S_{ij} - c^+}{\gamma_i^+}} + \sum_{k \in \mathcal{P}_i} e^{\frac{S_{ij} - S_{ik}}{\gamma_i^+}}},$$

$$p_{ij}^{-*} = \frac{e^{\frac{l_{ij}}{\gamma_i^-}}}{e^{\frac{l_{i,|\mathcal{N}_i|+1}}{\gamma_i^-}} + \sum_{k \in \mathcal{N}_i} e^{\frac{l_{ik}}{\gamma_i^-}}} \overset{y_{ij}=-1}{=} \frac{e^{\frac{[m+(S_{ij} - \lambda)]_+}{\gamma_i^-}}}{1 + \sum_{k \in \mathcal{P}_i} e^{\frac{[m+(S_{ik} - \lambda)]_+}{\gamma_i^-}}} = \frac{1}{e^{\frac{c^- - S_{ij}}{\gamma_i^-}} + \sum_{k \in \mathcal{N}_i} e^{\frac{S_{ik} - S_{ij}}{\gamma_i^-}}}, \tag{25}$$

where $c^+, c^-, \gamma_i^+, \gamma_i^-$ are hyperpatmeters. Similar to LS loss, waking the gradients to $\partial g(\theta, \mathbf{p}^*)$ in terms of $\theta$, we can get:

$$\frac{\partial g(\theta, \mathbf{p}^*)}{\partial \theta} = \sum_{i=1}^{B} \Big( \sum_{j \in \mathcal{P}_i} -p_{ij}^{+*} \frac{\partial S_{ij}}{\partial \theta} + \sum_{j \in \mathcal{N}_i} p_{ij}^{-*} \frac{\partial S_{ij}}{\partial \theta} \Big) \tag{26}$$

Similarly, we take gradients to the $\mathcal{L}_{MS}$ loss function (9):

$$\frac{\partial \mathcal{L}_{LS}}{\partial \theta} = \sum_{i=1}^{B} \Big( \sum_{j \in \mathcal{P}_i} \frac{-1}{e^{\alpha(S_{ij} - \lambda)} + \sum_{k \in \mathcal{P}_i}^{B} e^{\alpha(S_{ij} - S_{ik})}} \frac{\partial S_{ij}}{\partial \theta} + \sum_{j \in \mathcal{N}_i} \frac{1}{e^{\beta(\lambda - S_{ij})} + \sum_{k \in \mathcal{N}_i}^{B} e^{\beta(S_{ik} - S_{ij})}} \frac{\partial S_{ij}}{\partial \theta} \Big) \tag{27}$$

By substituting (25) into (21),and set $c^+ = \lambda + m, c^- = \lambda - m, \gamma_i^+ = \frac{1}{\alpha}, \gamma_i^- = \frac{1}{\beta}, i \in [B]$, it is obvious to show equation (21) and (27) are the same. As a result, our DRO framework also recovers the MS loss.

### 6.1.3 Recovering of HAP2S_E in Yu et al. (2018)

Yu et al. (2018) provides an a hardaware point to set (HAP2S) triplet loss with an adaptive hard mining scheme to address the sensitive issue caused by mining the hardest positive pair and negative pair in Hermans et al. (2017). The key of HAP2S loss is to assign different weights to the points in $\mathcal{P}_i$ and $\mathcal{N}_i$ for each anchor $\mathbf{x}_i$. We show that our DRO framework is able to recover HAP2S with exponential weighting scheme by exactly using the DRO formulation equation (7) for LS loss.

The triplet loss is defined as follows:

$$\mathcal{L}_{trp} = \frac{1}{N_t} \sum_{y_j = y_i, y_k \neq y_i} [S_{ik} - S_{ij} + m]_+ \tag{28}$$

where $[x]_+ = \max\{0, x\}$, $S_{ij}$ and $S_{ik}$ denote the similarity of positive pair $\{\mathbf{x}_i, x_j\}, j \in \mathcal{P}_i$, negative pair $\{\mathbf{x}_i, x_k\}, k \in \mathcal{N}_i$ for the same anchor $\mathbf{x}_i$, $N_t$ is the number of all possible triplets in the mini-batch. Without causing ambiguity, $S_{ij}^+ = S_{ij}$, when $j \in \mathcal{P}_i$, $S_{ij}^- = S_{ij}$, when $j \in \mathcal{N}_i$. Similar to pair losses, such as margin loss and binomial loss, there exist a huge amount of triplets in a batch that have no contribution to the $\mathcal{L}_{trp}$. As a result, pair mining is critical to improve the performance of the model.

Hermans et al. (2017) provide a variant of triplet loss by selecting the hardest positive pair and the hardest negative pair for each anchor. The formulation can be written as:

$$\mathcal{L}_{trpBH} = \frac{1}{B} \sum_{i=1}^{B} [\max_{j \in \mathcal{N}_i} S_{ij} - \min_{j \in \mathcal{P}_i} S_{ij} + m]_+ \tag{29}$$

The state-of-the-art results on two large-scale datasets has been reported based on this variant of triplet loss. However, equation (29) is sensitive to outliers which usually serve as the hardest sample. To increase the robustness of the model, HAP2S triplet loss has been proposed in Yu et al. (2018):

$$\mathcal{L}_{HAP2S} = \frac{1}{B} \sum_{a=1}^{B} [\mathcal{S}_{ij}^- - \mathcal{S}_{ij}^+ + m]_+, \tag{30}$$

where

$$\mathcal{S}_{ij}^- = \frac{\sum\limits_{k \in \mathcal{N}_i} q_{ij}^- S_{ij}^-}{\sum\limits_{j \in \mathcal{N}_i} q_{ij}^-}, \mathcal{S}_{ij}^+ = \frac{\sum\limits_{j \in \mathcal{P}_i} q_{ij}^+ S_{ij}^+}{\sum\limits_{j \in \mathcal{P}_i} q_{ij}^+}, \tag{31}$$

where $q_{ij}$ is the weights designed to each pair, for which they propose two variants of weighting schemes for HAP2S, i.e., exponential weighting and polynomial weighting. Here we show that our DRO formulation is able to recover the HAP2S with exponential weighting scheme (denoted by HAP2S_E), i.e. the weight $q_{ij}$ of each pair is an exponential function over its similarity:

$$q_{ij}^+ = \exp\left(\frac{-S_{ij}}{\gamma}\right), j \in \mathcal{P}_i, \quad q_{ij}^- = \exp\left(\frac{S_{ij}}{\gamma}\right), j \in \mathcal{N}_i. \tag{32}$$

Note that $q_{ij}^+$ and $q_{ij}^-$ are constant scalars that not involved in the gradient of $\mathcal{L}_{HAP2S}$ w.r.t to $\theta$.

Substitute equation (31) and (32) into (30)

$$\mathcal{L}_{HAP2S} = \frac{1}{B} \sum_{i=1}^{B} \left( \sum_{j \in \mathcal{P}_i} \left(\frac{-1}{\sum\limits_{k \in \mathcal{P}_i} \exp\left(\frac{S_{ij} - S_{ik}}{\gamma}\right)}\right) S_{ij} + \sum_{j \in \mathcal{N}_i} \left(\frac{1}{\sum\limits_{k \in \mathcal{N}_i} \exp\left(\frac{S_{ik} - S_{ij}}{\gamma}\right)}\right) S_{ij} \right) + D \tag{33}$$

where $D$ absorbs all constants.

Since $S_{ij} - S_{ik}$, $S_{ik} - S_{ij}$ in the exponential function is derived from $q_{ij}^+$ and $q_{ij}^-$, thus they are also a constant scalars when taking derivative to $\theta$. By taking derivative to $\mathcal{L}_{HAP2S}$, we get:

$$\frac{\partial \mathcal{L}_{HAP2S}}{\partial \theta} = \frac{1}{B} \sum_{i=1}^{B} \left( \sum_{j \in \mathcal{P}_i} \left( \frac{-1}{\sum_{k \in \mathcal{P}_i} \exp(\frac{S_{ij} - S_{ik}}{\gamma})} \right) \frac{\partial S_{ij}}{\partial \theta} + \sum_{j \in \mathcal{N}_i} \left( \frac{1}{\sum_{k \in \mathcal{N}_i} \exp(\frac{S_{ik} - S_{ij}}{\gamma})} \right) \frac{\partial S_{ij}}{\partial \theta} \right), \quad (34)$$

We can see that the subgradients of $\mathcal{L}_{HAP2E}$ in (34) are the same as the subgradients of LS loss in (22). As a result, $\mathcal{L}_{HAP2S}$ is a special case of our DRO framework.

## 6.2 SUPPLEMENTARY EXPERIMENTS

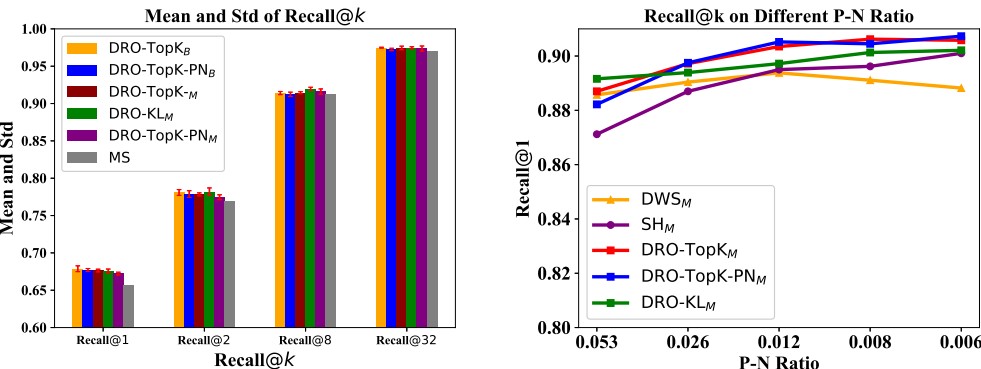

**Figure 4:** Mean and std of Recall@k over five runs comparing the best baseline performer (MS)

**Figure 5:** Recall@1 vs Imbalance Ratio on embedding space dimension 512

To investigate the effect of randomness of the stochastic algorithms and evaluate the robustness of our DRO framework, we report the average mean and standard variance of recall@$k$ on Cub-200-2011 in Figure 4. We do not plot recall@4 and recall@16 for better visualization. The experimental setting is the same as the experiments of the SOTA quantitative results we reported in Table 1 (section 4.1) but with five runs. The gray bars are the recall@$k$ of best performer among SOTA baselines, i.e., MS. It is clear to see that all our DRO variants outperform MS in terms of the average recall@$k$ over all different values of $k$. Specifically, the average recall@1 of DRO-TopK$_B$ is 67.9%, which has a significant improvement over the baselines, i.e., 65.7% of MS. In addition, the small standard deviation error bars imply that our DRO framework is robust enough to have a better performance than SOTA methods.

To show the effect of the network architecture to our DRO framework and its robustness, we additionally repeat the experiments of Recall/Imbalance Ratio in section 4.2.2, but with the embedding space dimensions 512 (rather than 1024 in Section 4.2.2). The results are illustrated in Figure 5. In comparison with the performance of recall@1 on embedding space dimensions 1024 in Figure 1, we can see that the fluctuations of recall@1 on different PN-Ratios are sublet when the feature embedding changes from 1024 to 512. For example, the recall@1 only changes, from 0.9046 to 0.9058 for DRO-TopK$_M$, from 0.9086 to 0.9073 for DRO-TopK-PN$_M$, from 0.9018 to 0.9021 for DRO-KL$_M$, when PN-Ratio is 0.006. Further, the trends of different methods imply that our DRO framework can consistently achieve competitive even best results compared with other SOTA methods (SH and DWS) under different embedding space dimensions. Since different P-N Ratios in corresponding to batchsizes $\{80, 160, 320, 480, 640\}$, the above experimental results verify that our methods are not too sensitive to the embedding space dimensions in different batchsizes, and also outperform other SOTA mining methods in different embedding space dimensions. In contrast, it shows that the baseline DWS is relatively more sensitive to the embedding space dimension. In Figure 5 where embedding space dimension is set to 512, DWS has a smaller performance drop from P-N ration = 0.012 to 0.006, while in Figure 1 where the embedding space dimension is 1024, DWS encounters a much larger performance decrease.

