# OpenReview forum: "A SIMPLE AND EFFECTIVE FRAMEWORK FOR PAIRWISE DEEP METRIC LEARNING"
_ICLR.cc/2020/Conference — Reject_

### Official Review · AnonReviewer2 · 2019-10-17
**Official Blind Review #2**

**Rating:** 6

**Review:**

This paper proposes a framework for deep metric learning. Using ideas from distributionally robust optimization, the loss (in each batch) is the worst case weighted average of all pairwise classification losses, taken over an uncertainty set of possible weights. The framework is shown to be general and encompass various previous approaches. Based on it, the authors propose several new algorithms, which are shown to outperform the SOTA on image retrieval data sets in terms of recall.

The main contribution of the paper is a unification of previous deep metric learning algorithms, which would be helpful to the community and could inspire new approaches. I found the empirical observation that the proposed algorithms are able to reduce the computation time by nearly half to be compelling. However, apart from DRO-TopK-PN, the proposed algorithms appear to be minor modifications of existing algorithms.

Questions about the experimental protocol:
1. Are the results from one run, or averaged over several? Standard errors of the evaluation metrics would be very helpful to judge the improvements made by the algorithms, especially as the algorithms are stochastic due to batching.
2. The proposed algorithms seem to be similar to those of Fan et al. (2017) and Namkoong and Duchi (2017). Is there a particular reason why they weren’t included in the experiments?

**Experience Assessment:**

I do not know much about this area.

**Review Assessment: Checking Correctness Of Derivations And Theory:**

I assessed the sensibility of the derivations and theory.

**Review Assessment: Checking Correctness Of Experiments:**

I assessed the sensibility of the experiments.

**Review Assessment: Thoroughness In Paper Reading:**

I read the paper at least twice and used my best judgement in assessing the paper.

---

> ### Author Response · Authors · 2019-11-14
> **Difference from traditional DRO**
>
> Thanks for your comments! For differences between our framework and traditional DRO method, please also check response to Reviewer 3. We want to emphasize that the modifications (i.e., defining over a mini-bath for the robust loss, and more general and flexible regularization of the dual variables) are subtle but very important for achieving better empirical results than complicated losses and bringing more theoretical insights for complicated losses.

---

### Official Review · AnonReviewer1 · 2019-10-21
**Official Blind Review #1**

**Rating:** 6

**Review:**

The authors address the (increasingly popular) problem of learning a metric from a given multi-dimensional data set. They consider the deep metric learning setup, where target distances are defined as the euclidean distances in an artificial feature space (created by a deep neural network). Main focus of the paper is to cases where the data set is affected by a substantial imbalance between the amount of examples that are similar to each other and the total number of examples.

I would tend to accept the paper because handling the imbalance problem in metric learning is important and both the theoretical analysis and the experiments show that the proposed method may have some impact.

The idea of reducing the problem to a binary classification between similar and dissimilar examples may look too simple but i) is a common approach in deep metric learning, ii) helps to handle the implicit imbalance problem and iii) suggests possible generalisations to other network-based problems (for example, where similarity is naturally defined by the existence of absence of a link). Showing that many complicated losses are equivalent to DRO may also help the general understanding of the metric learning task.

My main concerns are about the net contribution of the paper. Tackling the imbalance problem is important but it is not clear whether the full metric learning setup is really needed. The authors could have stated more precisely in what sense the metric learning unbalanced problem they consider is different from usual unbalanced binary classification. Otherwise, as DRO is well known, it is hard to identify the real novelty of their method.

Questions:
- how does the specific metric learning setup make the considered DRO different from usual unbalanced classification?
- how the network architecture affects the performance? For example, would the size of the embedding space change the recall/imbalance plot?
- Is the choice of euclidean distances standard in deep metric learning? Would a choice of more general distances be incorporated in the proposed method?



**Experience Assessment:**

I have read many papers in this area.

**Review Assessment: Checking Correctness Of Derivations And Theory:**

I did not assess the derivations or theory.

**Review Assessment: Checking Correctness Of Experiments:**

I assessed the sensibility of the experiments.

**Review Assessment: Thoroughness In Paper Reading:**

I read the paper at least twice and used my best judgement in assessing the paper.

---

### Official Review · AnonReviewer3 · 2019-10-23
**Official Blind Review #3**

**Rating:** 3

**Review:**

This paper casts deep metric learning (DML) as a pairwise binary classification problem such that pairs of examples need to be classified as similar or dissimilar. The authors propose an objective function that computes a weighted sum over the pairwise losses in a mini-batch. The weight vector is selected to maximize the objective from a decision set encoding constraints. This formulation is called the distributionally robust optimization (DRO) framework.

The authors argue that the DRO framework is theoretically justified by showing how certain decision sets result in existing machine learning loss functions. This portion of the paper seemed hand-wavy. It is not clear what is the purpose of including the theorem from Namkoon & Duchi. It would be more clear in my view to just make the short point that a certain decision set recovers the DRO with f-divergence as would be expected. The claims with regard to learning theory are over-stated in the paper.

The authors proposed three variants of the general framework. They include a top-K formulation, a variance-regularized version, and a top-K version using a balance between positive and negative examples. The DRO framework and the variants are the main contributions in terms of methodology in this paper. It is also shown that the framework generalizes more complicated recently proposed losses.

The experiments demonstrate the DRO framework consistently outperforms state of the art deep metric learning methods on benchmark datasets by small margins. There is also a computational speed advantage that is shown.
Overall, this paper shows that the ideas from distributionally robust optimization work well in deep metric learning. In particular, the paper shows that by combining the DRO framework with simple loss functions, performance comparable with complicated loss functions can be obtained. This aspect, along with the generality are the main strong suits. That being said, I do not see this paper to be that significant of a contribution. The main idea in the paper seems like a rather direct application of the DRO modeling framework and it does not provide too significant of improvement over the MS loss.  The paper was not written super clearly and was too long. Reviewers were instructed to apply a higher standard to papers in excess of 8 pages and this paper would have been presented more effectively if it was shorter. For these reasons, I recommended a weak reject.

**Experience Assessment:**

I have read many papers in this area.

**Review Assessment: Checking Correctness Of Derivations And Theory:**

I assessed the sensibility of the derivations and theory.

**Review Assessment: Checking Correctness Of Experiments:**

I assessed the sensibility of the experiments.

**Review Assessment: Thoroughness In Paper Reading:**

I read the paper at least twice and used my best judgement in assessing the paper.

---

> ### Author Response · Authors · 2019-11-14
> **Difference from traditional DRO framework  and our contributions to DML**
>
> We would emphasize that our framework is not a straightforward application of DRO. Instead, by addressing the critical issues in DML, our framework is an effective and general approach to DML. We summarize two significant contributions of our paper.
>
> First, our framework is more general, flexible and practical than traditional DRO. While traditional DRO usually restricts the dual variable to be on a simplex, our framework is built upon the minibatch and its uncertainty set for the dual variable is not necessarily restricted to such probability simplex. Please note that this is very important for us 1) to make the proposed approach practical for big data than traditional DRO defined on the whole data set; 2) to recover the approaches based on MS loss and LS loss by choosing different regularizations on the dual variables. It is notable that such recovery hinges on special grouping of the dual variables, which is not possible under traditional DRO framework; 3) to design more powerful variants such as DRO-TopK-PN, which is less sensitive to the positive to negative ratio (shown in Figure 1).
>
> Second, our framework introduces significant contributions to DML by 1) connecting loss development and sampling strategy design together, 2) justifying the existing loss functions in DML (e.g., MS and LS loss) from our general framework, and 3) providing insights to design new variants.
> In literature, as mentioned in our paper, many studies of DML either focus on developing increasingly complicated minibatch losses which rarely provide deep insights for why it is effective, or designing sampling strategies in terms of pairs. Consequently, developing loss functions and designing sampling strategies are two independent research directions, and may not benefit from each other. However, in our framework, we unify these two lines of work into a general framework to jointly take advantage of sampling and loss development. Specifically, the sampling weights according to the variable p is updated based on the relative magnitude of pairwise losses within a mini-batch at each iteration. Then the constructed robust loss is able to promote performance, which has been extensively demonstrated in our experiments.

---

> ### Author Response · Authors · 2019-11-14
> **Significant improvement over MS loss**
>
> We believe that our experimental improvement over MS loss is significant.
> Please note that (from Table 1 and 3), compared with the best baselines, MS improved 2.1% (over Margin), -1.1% (over ABE), 2.4% (over ABE) on Cub-200-2011, Cars-196 and In-Shop in Recall@1, respectively. On the other hand, our best DRO variant always achieves the best performance, improving 2.4% (over MS) on Cub-200-2011, 1.2% (over ABE), 2.5% (over MS) on Cars-196,  1.6% over MS on In-Shop. Among these variants of our framework, DRO-KL_M improves 2.0% (over MS), 2.5% (over MS), and 1.1% (over  MS) on Cub-200-2011, Cars-196 and In-Shop, respectively.
>
> In our ablation study, we show that DRO-KL-G could recover the performance of MS and LS by changing the hyper parameter \gamma (Table 2 and 4). Furthermore, tuning \gamma helps DRO-KL-G outperform MS in Recall@1 by 1.4%, 1.8% and 6% on Cub-200-2011, Cars-196 and In-Shop, respectively.

---

### Decision · Program_Chairs · 2019-12-19

**Decision:**

Reject

**Comment:**

The reviewers agree that this is a reasonable paper but somewhat derivative. The authors discussed the contribution further in the rebuttal, but even in light of their comments, I consider the significance of this work too low for acceptance.